



# Geospatial dataset for Hydrologic analyses in India (GHI): A quality controlled dataset on river gauges, catchment boundaries and hydrometeorological time series

Gopi Goteti[1]

[1]5741 NW 92nd Ct, Johnston, Iowa, 50131, USA

**Correspondence:** Gopi Goteti (saagu.neeru@gmail.com)

**Abstract.**

Streamflow gauging stations not only track the pulse of rivers but also act as common reference points for hydrologic and other environmental analyses. As such, streamflow data and metadata on gauging stations - GIS data on station locations, their upstream catchment boundaries and river flow networks, are critical for analyses. However, for India's river basins, availability of such data is limited; when available, data is not in an analysis-ready format and can have substantial errors. Studies often use available information from India's water agencies as is, without checking for its validity. This study addresses the above limitations by building a new dataset using existing metadata (from CWC and WRIS) and checking it against publicly available information from global data sources (e.g., WWF, MERIT and Copernicus), and online maps (e.g., Google Maps). The quality control process categorizes existing metadata based on its consistency with these sources; also, existing metadata is supplemented with additional information where needed. The new dataset developed here is called the 'Geospatial dataset for Hydrologic analyses in India' (GHI) and uses HydroSHEDS data as the underlying template. GHI has both geospatial and time series information. In this initial version of GHI, the spatial domain includes only the river basins of Peninsular India where daily streamflow data is publicly available.

Following the quality control process, CWC's 645 stations in Peninsular India were categorized into three groups - Group 1 (reliable metadata, adequate daily streamflow data; 213 stations), Group 2 (reliable metadata, inadequate or no daily streamflow data; 259 stations), and Group 3 (missing or unreliable metadata; 173 stations). For each of the 472 stations falling in Groups 1 and 2, catchment-specific annual and monthly time series spanning 71 water years (1950-2020) of the following were compiled: observed precipitation from IMD, observed streamflow from WRIS, estimated precipitation, evapotranspiration (ET) and streamflow from ERA5-Land, and ET from GLEAM. A preliminary analysis of catchment-scale time series data indicates that while the compiled data appears reasonable over most of the study domain, spurious runoff-precipitation ratios were observed in the hilly coastal regions of Western India. This adds to yet another data-related obstacle faced by the hydrologic community. In order to quantify historical changes and reconcile them with anticipated future changes, the community needs robust and reliable hydrographic and hydrometeorological datasets, as well as unrestricted access to such datasets. The goal of this study is to highlight the limitations of existing datasets and pave the way for a community-led effort towards building the needed datasets. GHI serves as a placeholder until such datasets become available. Potential improvements to GHI are discussed.



# 1 Introduction

Water resources assessments and other large-scale hydrologic analyses are important and useful means to objectively quantify the water budget. Such analyses often require reconciling observed streamflow at a gauging station with estimated or modeled fluxes over the upstream catchment area. As such, having accurate GIS data on gauging station locations and catchment boundaries is critical. However, for India's watersheds, such metadata - GIS data on river gauging station locations, river networks and catchment area boundaries, etc., is publicly available only to a limited extent. The primary sources of such information are the Central Water Commission (CWC) and the online Water Resources Information System (WRIS, https://indiawris.gov.in/wris). Data from CWC, if available, is buried within its various reports and users need to piece together the needed information from such reports. WRIS addresses some of CWC's deficiencies. However, catchment area boundaries from WRIS are available only for the large river basins. Contributing catchment areas outside of India are excluded by WRIS.

There are further data-related challenges when it comes to hydrologic analyses over India. Streamflow data is available through WRIS only if the river basins are entirely within India's boundaries. Hence, WRIS data is only available for the basins of Peninsular India (shaded regions in Figure 1, panel (a)). For river basins such as Ganga, Brahmaputra and Indus, which have catchment areas spanning multiple countries, data is 'classified' by CWC and is not publicly available (non-shaded regions in Figure 1). Thus, streamflow data for a large portion of India is not readily available. Analysts could use compiled information from established sources such as the Global Data Runoff Center (GRDC, https://www.bafg.de/GRDC/), the Global Monthly River Discharge Data Set (RivDIS, Vorosmarty et al. (1998)), Global Streamflow Indices and Metadata Archive (GSIM, Do et al. (2018), Gudmundsson et al. (2018)) or other global databases. RivDIS and the GRDC contain legacy data only for a small fraction of the gauging stations currently operated by CWC. There is limited or no information available on the specific Indian entities which supplied this data to GRDC or RivDIS, or the extent of missing streamflow data in these sources. GSIM's streamflow data for India is based solely on 'non-classified' information already available from WRIS.

Due to a lack of readily available metadata or hydrographic data, studies often compile the needed information, using whatever is available from CWC and WRIS as a reference. For instance, Shah and Mishra (2016) calibrated and validated their hydrologic model at 18 stations across India, most of which were from CWC/WRIS and the rest from RivDIS. It is not known whether Shah and Mishra (2016) accounted for any of the missing streamflow data in their study. Madhusoodhanan et al. (2017) assessed uncertainties in modeled fluxes using 20 stations from WRIS whose streamflow was minimally affected by human interventions. The station metadata from WRIS was used as is without checking for its validity. CWC used more than a hundred stations across India within their water resources assessment (CWC-19 (2019)), but did not provide any GIS data on the stations and their catchment boundaries. Goteti (2022) noted some discrepancies in estimated catchment areas from CWC-19 (2019).

Ganguli et al. (2022) used catchment boundaries compiled by GSIM to quantify hydrometeorological variables of interest in their analysis of drought in Peninsular India. GSIM's goals were similar to those of this study but for thousands of stations



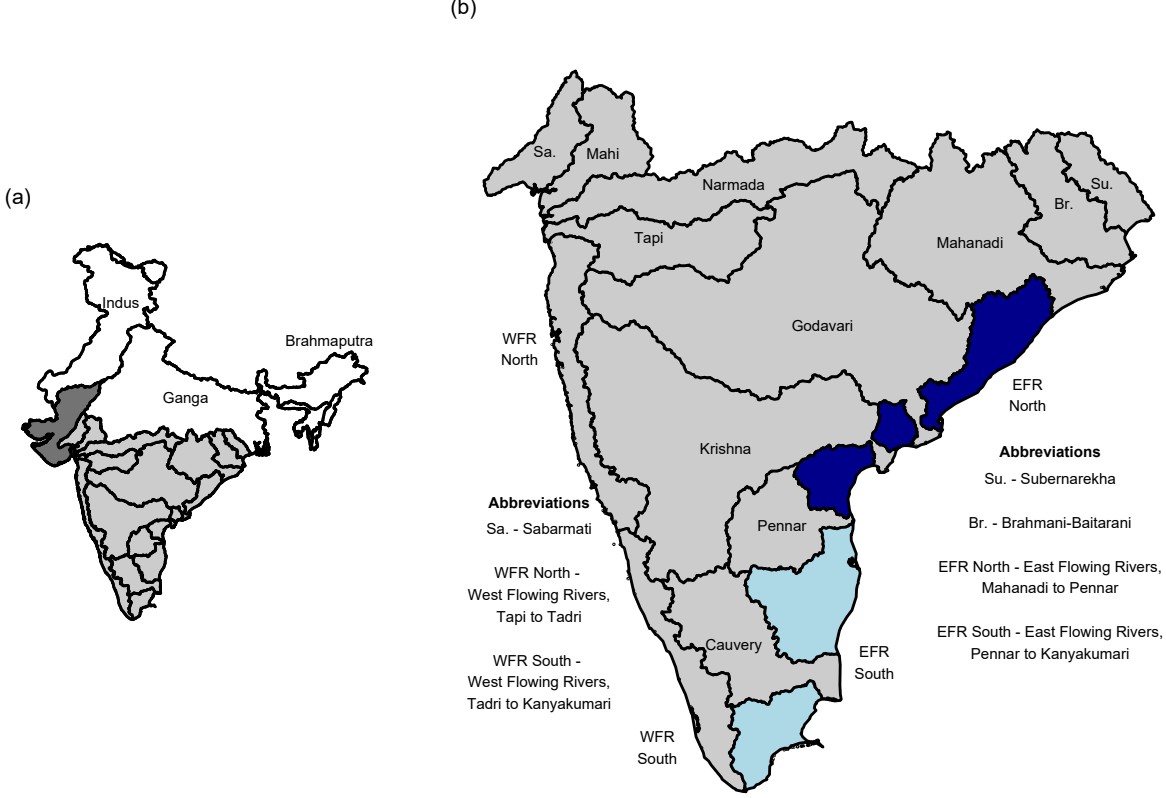

**Figure 1.** (a) Major river basins of India and those used in this study (lightly shaded). (b) Composite basins of Peninsular India identified within GHI. Some basin names within the map are abbreviated for ease of display but elaborated next to the map.

across may countries (Do et al. (2018), Gudmundsson et al. (2018)). As such, GSIM could not perform manual verification of
available station metadata. GSIM tailored its boundary delineation process by relocating the stations on to the river network
such that the final catchment area estimates matched the 'reference' catchment area estimates from CWC. Catchment bound-
aries were derived for select stations available from CWC's inventory from 2012. While relocating stations on to river networks
to match the 'reference' catchment area is a reasonable approach (and is also used by GRDC and other studies), the 'reference'
metadata from CWC or WRIS has several limitations and is not completely reliable. Station locations are prone to substantial
errors, catchment area estimates are inconsistent with other sources (and sometimes inconsistent with CWC's own reports).
The following is a discussion on these limitations.



## 1.1 Issues with Existing Data

### 1.1.1 Spurious Station Locations

Actual stations locations can sometimes be hundreds of kilometers away from their current location based on CWC's published
coordinates. Some such examples are identified in Figure 2 and Table 1. In some instances, current station locations fall in the
Bay of Bengal (Station S2 in Table 1) or in the Arabian Sea (Station S5 in Table 1).

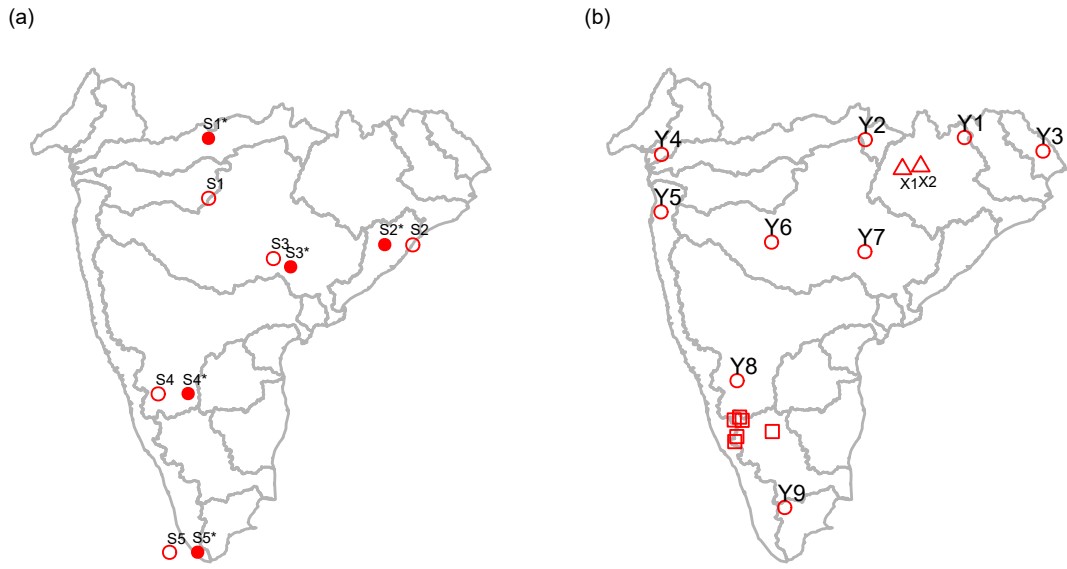

**Figure 2.** Select stations used as examples to illustrate issues with existing metadata. (a) Stations discussed in Table 1. The current locations from CWC are shown as open red circles, while the potential correct locations are shown as shaded red circles; (b) Stations discussed in Table 2 (squares), Table 3 (triangles) and Table 4 (circles).

    One particular example is further illustrated in Figure 3. According to CWC, the Eturunagaram station is on the Godavari
river. However, when overlaid on Google Maps, the current location falls on the Manair river, near the Manair Bridge. More-
over, the current location of Eturunagaram almost coincides with the current location of another CWC station (Somanpally
station). The potential correct location is more than 60 km to the South-East, by the town of Eturunagaram, close to the Go-
davari river. Based on the visual pattern of current locations and their potential correct locations in Figure 2 (panel (a)), it
appears that these errors could be due to typographical errors in station coordinates - the potential correct locations are gen-
erally along the latitude or the longitude passing through the current location. As such, the station coordinates from CWC are
not reliable, and they individually need to be verified with other sources, such as Google Maps, before one can proceed with
any analysis.



**Table 1.** Examples of stations with spurious locations, labeled S1 through S5 in Figure 2, panel (a). Catchment area is in $\mathrm{km}^2$.

| ID | CWC ID | Site Name | River/Basin | Catchment Area |
|----|--------|-----------|-------------|----------------|
| S1 | CW1NAM001443 | Awalighat | Narmada | 45,598 |
| S2 | CW1VAM000996 | Sirjholi | Sirjoli Nala | 460 |
| S3 | CW1PRA000667 | Eturunagaram | Godavari | 270,600 |
| S4 | CW1TUL000762 | Hoovinahole | Krishna | 2,585 |
| S5 | CW1PAR000447 | Aruvipuram | Neyyar | 194 |

### 1.1.2 Spurious Catchment Area Estimates

Drawbacks of catchment area estimates from CWC are illustrated using examples in Tables 2, 3 and 4. In some cases, the catchment areas for different stations in the basin are identical - for instance, the six stations in the Cauvery basin identified in Table 2, have the identical catchment area of 6,410 $\mathrm{km}^2$. In some cases, the catchment area for upstream stations is higher than

the catchment area for the downstream station. For instance, Simga station in the Mahanadi basin is upstream of Jondhra and should have a smaller catchment area, but the opposite is the case - 30,761 $\mathrm{km}^2$ at Simga versus 29,645 $\mathrm{km}^2$ at Jondhra (Table 3). The catchment area estimates from this study, discussed in the subsequent Sections, do not have such issues.

**Table 2.** Catchment area estimates from CWC for select stations in the Cauvery basin. The locations of these stations are shown as squares in Figure 2, panel (b). Catchment area is in $\mathrm{km}^2$

| CWC ID | Site Name | River/Basin | Catchment Area |
|--------|-----------|-------------|----------------|
| CW1CAU001108 | Beluru | Hemavathy | 6,410 |
| CW1CAU000978 | Bettadamane | Cauvery | 6,410 |
| CW1CAU001188 | Jannapura | Cauvery | 6,410 |
| CW1CAU001271 | Mukkkodlu | Cauvery | 6,410 |
| CW1CAU000906 | Napoklu | Cauvery | 6,410 |
| CW1CAM001272 | Thoreshettyhalli | Cauvery | 6,410 |

**Table 3.** Potential erroneous catchment area ($\mathrm{km}^2$) from CWC for Simga station in the Mahanadi basin. The corresponding values from GHI and MERIT (estimated in Section 3) are also shown. The ID for each station corresponds to its location in Figure 2.

| ID | CWC ID | Site Name | River/Basin | CWC | GHI | MERIT |
|----|--------|-----------|-------------|-----|-----|-------|
| X1 | CW1MAU000247 | Simga | Mahanadi | 30,761 | 16,903 | 16,803 |
| X2 | CW1MAU000515 | Jondhra | Mahanadi | 29,645 | 29,822 | 29,620 |

Earth System
Science
Data

**Figure 3.** (a) Erroneous station location from CWC for the Eturunagaram station in the Godavari basin. The potential correct location is at least 60 km southeast of CWC's current location and is verified using Google Maps - black square indicates a reference landmark corresponding to the town of Eturunagaram. (b) CWC's current location for the Eturunagaram station suspiciously coincides with CWC's location for the Somanpally station in the Godavari basin. (c) The potential correct location for the Eturunagaram station is on the Godavari River, by the town of Eturunagaram, matching CWC's site description of the station.

Another issue with the catchment areas from CWC is the extent of rounding used in the presentation of these estimates. Table 4 shows nine select stations in various basins whose catchment area varies from about 1,000 $\mathrm{km}^2$ to about 40,000 $\mathrm{km}^2$. Based on these catchment areas, those presented in Tables 1, 2 and 3, and a general examination of other CWC estimates, the rounding used by CWC appears to be arbitrary. While in some cases the rounding used by CWC can be reasonable (e.g., Jamsholaghat



station, Y3), in other cases it can result in large departures from actual values (station Y7 and Y9) or be completely unreasonable (station Y1).

**Table 4.** Select catchment area ($km^2$) estimates from CWC to showcase the rounding used by CWC. The ID for each station corresponds to its location in Figure 2, panel (b).

| ID | CWC ID | Site Name | River/Basin | CWC | GHI | MERIT |
|----|--------|-----------|-------------|-----|-----|-------|
| Y1 | CW1MAM000657 | Thettatanagar | Mahanadi | 2,500 | 1,495 | 1,461 |
| Y2 | CW1NAU000327 | Mandla | Narmada | 13,000 | 12,944 | 12,919 |
| Y3 | CW1SUB000352 | Jamsholaghat | Subarnarekha | 16,000 | 15,979 | 15,821 |
| Y4 | CW1MHL000680 | Pingalwada | Dhadhar (Independent river) | 2,400 | 2,566 | 2,426 |
| Y5 | CW1BHT000681 | Madhuban Dam | Damanganga | 1,800 | 1,865 | 1,817 |
| Y6 | CW1GDM000115 | Purna | Godavari | 15,000 | 15,555 | 15,492 |
| Y7 | CW1IND000089 | Pathagudem | Godavari | 40,000 | 38,980 | 38,774 |
| Y8 | CW1TUU000620 | Byaladahalli | Krishna | 2,300 | 2,508 | 2,445 |
| Y9 | CW1PAM000468 | Theni | Vaigai | 1,200 | 1,364 | 1,319 |

### 1.1.3 Other Issues

As mentioned earlier, daily streamflow data is available through WRIS only for stations in Peninsular India (WRIS-OL (2022)). Such data can have missing data. While seasonal stations have data available only for a portion of the year, perennial stations can sometimes have a number of days with missing observations (i.e., blanks in the raw data). Ignoring such values when estimating monthly or annual statistics could lead to underestimation of aggregate streamflow and potential misrepresentation of the regional water balance.

Daily, monthly and annual streamflow data from WRIS-OL can be downloaded as spreadsheets. Within these spreadsheets, river gauging station names, parent river, parent river basin, and other relevant information is provided. However, there is no use of the station identification codes developed by CWC. It appears that WRIS contains streamflow data not only corresponding to CWC stations but also stations from other agencies (such as state and regional agencies). As such, the user needs to manually match the WRIS stations to the CWC stations based on available station description information. Given WRIS

contains duplicates and sometimes conflicting information on some stations, the user is burdened to infer the streamflow data corresponding the desired station and check for missing values, and fill-in for missing values. There appears to be at least a few years of latency in the data provided by WRIS. Data for the current season is not available. The individual values within the downloaded spreadsheets do not have any quality control flags associated with them - such as those indicating overbank flow, gauge malfunction, outlier data, etc.



## 1.2 Motivation for this Study

In the presence of the above discussed limitations, the analysts have to rely on their individual abilities to clean the data and compile the needed information. The lack of reliable metadata on river gauges and catchment boundaries affects the compilation, and subsequent analysis, of catchment-scale hydrometeorological variables. India's river basins have witnessed a rise in average temperature, a decrease in monsoon precipitation, a rise in droughts, along with a number of other hydro-climatological changes (Krishnan et al. (2020)). In order to better understand and quantify such changes, the research community needs robust hydrometeorological datasets across all river basins of India. Reliable hydrographic data is a fundamental building block of such datasets. Increasingly available satellite and other high-resolution data products can be leveraged to build the needed datasets. Moreover, high-resolution river discharge measurements from the new Surface Water and Ocean Topography mission (SWOT, https://swot.jpl.nasa.gov/) can be reconciled with historical information only when there is robust hydrographic data. Leveraging the state-of-the-art remote sensing data to build the needed hydrographic datasets is the motivation behind this study.

The National Hydrography Dataset (NHD, https://www.usgs.gov/national-hydrography), from the US Geological Survey (USGS) is an excellent example of a robust hydrography dataset. It is not feasible for one individual to build such a dataset for India's river basins, and requires agency-level or community-wide effort. Based on the recent reports available from WRIS, it appears that WRIS is progressing towards creating such a dataset. In the interim, hydrologists and other analysts still need a reliable dataset on India's river gauging stations, catchment boundaries and other relevant data. By making use of the widely used HydroSHEDS dataset as the underlying template, this study leverages the valuable and abundant data resources available via HydroSHEDS. The specific goal of this study is to build a quality-controlled analysis-ready dataset from publicly available resources for use in hydrologic analyses while highlighting the major limitations of existing datasets.

The remainder of this paper is organized as follows. In Section 2 the data sources used in this study are discussed. In Section 3 the methodology used to differentiate the more reliable gauging stations from the less reliable stations is described. The final product from this study, called the Geospatial dataset for Hydrologic analyses in India (GHI), is described in Section 4. In Section 5, a preliminary analysis of GHI's time series data is discussed. In Section 6 is a discussion on the limitation of GHI and potential next steps. The conclusions from this study are presented in Section 8.

## 2 Data

A number of datasets were used in this study and they are described in this Section and outlined in Table 5. These datasets pertain to metadata on river gauging stations, publicly available hydrography products, online maps, and observed or modeled hydrometeorological data. The two main agencies providing metadata on India's river gauging stations and streamflow data are CWC and WRIS. Several reports from these agencies were used in this study and the following notation is used to distinguish between the various reports from these agencies. CWC-21: list of latest active streamflow gauging stations as of 2021 (CWC-21, 2021); CWC-19: Water resources assessment by CWC and WRIS conducted in 2019 (CWC-19, 2019); CWC-YB: annual yearbooks published by CWC which contain statistical summaries on select streamflow gauging stations (CWC-YB, 2021);




**Table 5.** Datasets used in this study and relevant information.

| Data Type | Raw Format | Source | Purpose | Vintage |
|---|---|---|---|---|
| Station Metadata | PDF[a] | CWC-21 | used within GHI | Sep. 2021 |
| Station Metadata | Spreadsheet | WRIS-OL | Comparison w/ GHI, CWC | Jul. 2022 |
| Hydrography | GIS[b] | HydroSHEDS | used within GHI | 2020 |
| Hydrography | GIS[b] | MERIT | Comparison w/ GHI, CWC | Jul. 2022 |
| Composite River Basin Boundaries | GIS[b] | WRIS-GIS | Comparison w/ GHI | Dec. 2021 |
| River Network | GIS[b] | Lin et al. (2021) | Comparison w/ GHI | Dec. 2021 |
| Online Maps | - | Google Maps and OpenStreetMap | Verification of station locations and river networks | 2022 |
| Observed Streamflow | Spreadsheet | WRIS-OL | used within GHI | Aug. 2022 |
| Observed Streamflow | PDF[a] | CWC-YB | Comparison w/ WRIS-OL | Dec. 2020 |
| Precipitation | NetCDF[c] | IMD | used within GHI | May 2022 |
| Modeled Estimates (Precip., ET and runoff) | NetCDF[c] | ERA5-Land | used within GHI | Oct. 2022 |
| ET | NetCDF[c] | GLEAM | used within GHI | Jun. 2022 |

[a] PDF is portable document format; [b] GIS includes a variety of formats such as shapefiles, raster files and geodatabases; [c] NetCDF is a file format often used to store and share gridded meteorological data.

WRIS-GIS: GIS data on major river basin boundaries used by WRIS, obtained from WRIS via data request (WRIS-GIS, 2021); WRIS-OL: the online system from WRIS to disseminate streamflow and other hydrological data (WRIS-OL, 2022); WRIS-BR: Basin reports published by WRIS in 2014 (WRIS-BR, 2014).

## 2.1 Station Metadata

### 2.1.1 CWC

India's river gauging stations are maintained by CWC as well as regional agencies (Chatterjee and Sinha (2014)). However, metadata on these gauging stations is limited to stations maintained by CWC and is available via a number of reports from CWC and WRIS. For instance, the annual year books published by CWC (CWC-YB) contain information on station location and also streamflow measurements for select stations. The basin reports published by WRIS (WRIS-BR) contain maps of river
basins along with the location of select gauging stations. However, the underlying GIS data on basin boundaries and other relevant information is not available from WRIS (personal communication, January 4, 2022). The only publication from CWC which contains the latest information on all of its active river gauging stations is in the form of a PDF file from 2021 (CWC-21).

This PDF file was used in this study as the definitive source of station metadata from CWC.

### 2.1.2   WRIS

The online WRIS portal (WRIS-OL) provides station coordinates within the downloadable streamflow data files. A snippet of this data is shown in Figure A1 in Appendix A. A preliminary analysis indicated that in some instances station coordinates from WRIS-OL matched those from CWC-21, but there were many instances where blatant errors were found, such as coordinates

falling in the southern hemisphere or a location with the same name having differing coordinate values. Due to these reasons, station coordinates from WRIS-OL were not used in this study.

### 2.2   Hydrography

#### 2.2.1   Major River Basin Boundaries

CWC and other water agencies subdivide the entire country into major basins for administrative and data management pur-

poses. Due to the presence of river deltas, coastal basins and islands, and due to national boundaries not always following topography, river basins are conveniently lumped to create 'composite' river basins. CWC often uses such composite basins to cross-reference its data. The online WRIS portal (WRIS-OL) also uses similar composite basins from CWC to reference the stations associated with its streamflow data. However, other water agencies of India appear to have developed their own composite basins (https://indiawris.gov.in/wiki/doku.php?id=river_basins). This study also uses such composite basins for ease

of illustration and cross-referencing (see Section 3.1).

#### 2.2.2   HydroSHEDS

HydroSHEDS (Hydrological data and maps based on Shuttle Elevation Derivatives at multiple Scales, Lehner and Grill (2013), Lehner et al. (2008)) is one of the few high-resolution, global, quality-controlled and analysis-ready datasets currently available to the scientific community. It has been widely used by a number of studies worldwide and is continually being improved.

HydroSHEDS version 1, used in the creation of GHI, is derived primarily from Shuttle Radar Topography Mission (SRTM) elevation data at 3 arc-second resolution (about 90 m at the equator).

The suite of products from HydroSHEDS are the basis of the geospatial information within GHI. The core products from HydroSHEDS are raster datasets such as the digital elevation model (DEM) and derived flow direction. The secondary products from HydroSHEDS are derived from the core products and include vector products such as watershed boundaries (Hy-

droBASINS) and river networks (HydroRIVERS). This study makes extensive use of HydroBASINS and HydroRIVERS. HydroBASINS is a series of vectorized polygon layers on watershed boundaries nested within larger river basins. Within HydroBASINS, the Pfafstetter coding system is used to represent the hierarchical nesting of watersheds from level 1 (PF-1, large

river basins) through level 12 (PF-12, smallest watersheds). A sample graphic illustrating watersheds at various PF levels is in Appendix A (Figure A2). HydroRIVERS is a vectorized line network of river paths where the upstream catchment area is at least 10 km$^2$ or where the estimated average streamflow is at least 0.1 m$^3$/s, or both. Both HydroBASINS and HydroRIVERS are derived from the core HydroSHEDS products after aggregating to a 15 arc-second resolution (about 500 m at the equator). Thus, the underlying resolution for spatial products within GHI is 500 m.

### 2.2.3 MERIT

While several products similar to HydroSHEDS are available - such as HDMA (Verdin (2017)), MERIT (Yamazaki et al. (2019)), river networks (Yan et al. (2019)) and river drainage density (Princeton, Lin et al. (2021)), such products neither have the breadth of data nor the widespread usage HydroSHEDS currently has. MERIT-based products include relatively fewer sub-datasets, but MERIT-based products continue to grow. MERIT was developed at 3 arc-second resolution (about 90 m at the equator) after correcting for errors using multiple satellite datasets and filtering techniques. While upstream catchment area is readily available from MERIT, river networks are not. However, Lin et al. (2021) used MERIT to delineate high-resolution river networks at a global scale. This river network from Lin et al. (2021) was used in this study as a proxy for river network data from MERIT. The information from MERIT serves as an independent check on the HydroSHEDS data.

### 2.3 Online Maps

Landmarks such as rivers, bridges and highways are often used as reference points by CWC to describe the location of its gauging stations. Since publicly available maps from Google (Google Maps) and OpenStreetMap (OSM) contain such landmarks, they can be used to verify CWC's station description. The names of nearby towns and cities are also available from these sources and serve as reference landmarks. Such landmarks can be independently verified by users and are useful in the validation of station coordinates and other metadata. Google Maps typically contain more information on cities and towns than OSM. However, OSM typically has more information than Google Maps on rivers and other water bodies. An example graphic illustrating some of these differences between Google Maps and OSM is shown in Figure A3, Appendix A. The publicly available QGIS software (https://qgis.org/en/site/) has plug-ins for both Google Maps and OSM. QGIS was used throughout this study for GIS analyses and the plug-ins for online maps were used wherever needed.

### 2.4 Streamflow

### 2.4.1 WRIS

Daily runoff data from WRIS-OL was used to estimate monthly and annual runoff as needed. WRIS provides daily river stage and/or daily river flow information for gauging stations. WRIS enforces a limit of one calendar year's worth of data for each download. Thus, the user has to go through the tedious process of downloading one year at a time. Such data was downloaded for more than 70 years for all available stations. Only daily river flow data was used here. River stage information was not used





since appropriate stage-discharge relationships were not readily available. A sample of raw data from WRIS is shown in Figure A1 in Appendix A.

### 2.4.2   Other Sources

Streamflow data is available through global databases but was not used in this study since there is only limited information on the quality of such datasets. Only a brief description is provided here to make the reader aware of their existence. GRDC contains legacy streamflow data for only a handful of stations in India. Moreover, usage of data from the GRDC is permitted only after approval of a written request. It is also not known which specific Indian agency (CWC or other) provided data on India's gauging stations to GRDC and the quality of such data. RivDIS contains monthly discharge data for 1,018 stations around the world, including some in the GHI domain. However, this database has not been updated in more than 20 years. Moreover, the source of the streamflow data for India's gauging stations and the quality of such data is unknown. Due to these reasons, data from neither GRDC nor RivDIS was used in this study. GSIM contains streamflow data at gauging stations, among other metrics for use in hydrologic analyses. GSIM's streamflow data for India is based solely on information available from WRIS from around the year 2017. Thus, GSIM does not have any information not already available for India's river basins, and was not used in this study.

### 2.5   Hydrometeorological Data

Long-term hydrometeorological products spanning the entire study domain, often cited in scientific literature as reasonable products for use in hydrologic analyses, and currently being maintained and/or developed were chosen for this study. Gridded precipitation dataset from Indian Meteorological Department (IMD, Pai et al. (2014)) has become the benchmark precipitation dataset and has been used by a number of studies (e.g., Rana et al. (2015), Rani et al. (2021), Thakur et al. (2019)). Mahto and Mishra (2019) evaluated ERA5 products (ERA5 is the predecessor of ERA5-Land) and found that ERA5 is superior to other reanalysis products analyzed in their study, and is suitable for hydrologic assessments over India. Goteti (2022) used ET from ERA5-Land and GLEAM, among other products, for water resources assessment in the Godavari and Krishna river basins. This study specifically uses data from IMD, ERA5-Land and GLEAM. Sample precipitation and ET data for WY 2020 is shown in Figure A4 in Appendix A.

### 2.5.1   IMD

The IMD dataset used here is the monthly total precipitation for the period 1950-2020 on a 0.25° lat-lon grid (about 25 km at the equator).

### 2.5.2   ERA5-Land

ERA5-Land is based on the land component of the European Centre for Medium-Range Weather Forecasts (ECMWF) ERA5 climate reanalysis (Muñoz-Sabater et al. (2021)). The dataset used here is the post-processed monthly data on a 0.10° lat-lon



grid (about 10 km at the equator) for the period 1950-2020. In the remainder of this paper ERA5-Land is referred to as ERA5, unless otherwise specified.

### 2.5.3 GLEAM

Global Land Evaporation Amsterdam Model (GLEAM) is a set of algorithms that separately estimate the different components of ET, including transpiration, bare-soil evaporation, interception loss, open-water evaporation and sublimation (Martens et al. (2017), Miralles et al. (2011)). The dataset used here is the GLEAM v3.6a global dataset (https://www.gleam.eu/) which provides monthly total ET for the period 1980-2020 on a 0.25° lat-lon grid (about 25 km at the equator).

## 3 Methodology

GHI contains both geospatial and time series data. The geospatial component of GHI includes three layers, and the time series component includes annual and monthly observations or modeled estimates of hydrometeorological variables such as precipitation, ET and runoff. The end product includes the above information in typical data sharing formats such as shapefiles and plain text files. An overview of GHI is presented in Figure 4. Specific quality control measures followed within the creation each component of GHI are described in Figure 7. The remainder of this Section describes the creation of these individual components of GHI.

### 3.1 Composite Basins

The study domain comprises of only those river basins of India where daily streamflow data is publicly available (Figure 1). For ease of analysis and to be consistent with WRIS-GIS, the study domain was separated into composite river basins. Such composite basins serve as a regional spatial reference. Every gauging station within GHI is tagged with the parent composite basin information. Starting with PF-12 watersheds from HydroSHEDS, composite basins were manually delineated using the QGIS software. It was ensured that the boundaries of these basins followed topographic divides and the resulting basins were consistent with those from WRIS-GIS to the extent possible. The study domain was grouped into 15 composite hydrologic basins (Figure 1). The names of these composite basins are generally consistent with those from WRIS-GIS, but more consistent with those from CWC-19. Most composite basins are contiguous and are shown in grey shading in Figure 1. Basins that are not contiguous are shaded, and include the EFR North basin (dark blue), and the EFR South basin (light blue).

The catchment area of these GHI composite basins is compared with equivalent basins from other sources in Table 6. The categorization of basins is not the same across sources, and hence, a new category ('WFR North & South') was created to facilitate comparison. For ease of comparison, the different sources are compared against WRIS-GIS. The discrepancies between the different sources are typically less than 5%. GHI estimated catchment areas are much more closer to CWC-19 than to WRIS-GIS. Discrepancies between the various estimates are attributed to differences in the underlying topographic data's spatial resolution, land versus ocean demarcation used within the data and GIS projection system and/or coordinate system used.

**GIS layer 1: Composite Hydrologic Regions**

Major river basins based on HydroSHEDS; make boundaries consistent with WRIS boundaries

**GIS layer 2: Streamflow Stations**

Verify station locations; add reference landmarks

Relocate stations on to HydroSHEDS river network and MERIT Hydro network

**GIS layer 3: Station-specific Upstream Catchment Boundary & River Network**

Identify upstream catchment area boundary and river network for each station

Calculate catchment areas

**Observed Streamflow Data:** **2**

Compile monthly and annual time series from daily data; check for missing data (see Section 3.4)

**Quality Control Measures:** **3**

See Figure 7;

Categorize stations into Group 1, Group 2 and Group 3

**Compile Time Series Data:** **4**

Gridded data: identify grids falling within catchment boundary of each station

Hydrometeorological data: estimate grid area-weighted precip., ET and runoff for each catchment; add observed streamflow data; generate monthly and annual time series

**GHI final product:** **5**

Plain text files: station metadata, hydrometeorological time series data

Shapefiles (GIS): Composite river basins, station metadata (station raw locations, relocated locations, landmarks, other attributes), station-specific catchment boundaries and station-specific river networks

PDF files: one-page summary graphic on each station; monthly and annual summaries

**Figure 4.** Overview of GHI's data components.

Within GIS analyses, it is customary to choose an appropriate coordinate projection system which determines how three di-
mensional landscapes can be projected on to two dimensions for analysis and visualization. Estimation of area on the surface of the earth, such as catchment area, is affected by the choice of the projection system or coordinate system and is based on a number of factors (e.g., https://pro.arcgis.com/en/pro-app/latest/help/mapping/properties/coordinate-systems-and-projections.htm). While the basin boundaries from WRIS-GIS are in the Lambert Conformal Conic projection (https://pro.arcgis.com/en/pro-app/latest/help/mapping/properties/lambert-conformal-conic.htm), it is not known whether this is the best or recommended
projection system for India's river basins. Data sources such as MERIT estimate catchment areas using the WGS84 coordinate reference system (https://pro.arcgis.com/en/pro-app/latest/help/mapping/properties/specify-a-coordinate-system.htm), without using a specific projection system. Gridded meteorological data often use the WGS84 coordinate reference system as well. For the sake of convenience, this study also uses the WGS84 coordinate reference system.



**Table 6.** Summary of catchment area ($\mathrm{km}^2$) by composite basin from various sources. The numbers in parentheses indicate the percentage deviation from WRIS-GIS values.

| Composite Basin | WRIS-GIS | CWC-19 | GHI |
|---|---|---|---|
| Brahmani & Baitarani | 51,897 | 53,902 (4%) | 54,890 (6%) |
| Cauvery | 85,626 | 85,167 (-1%) | 85,361 (0%) |
| EFR North | 79,920 | 82,073 (3%) | 82,680 (3%) |
| EFR South | 102,288 | 101,657 (-1%) | 102,082 (0%) |
| Godavari | 302,063 | 312,150 (3%) | 311,732 (3%) |
| Krishna | 254,750 | 259,439 (2%) | 260,565 (2%) |
| Mahanadi | 139,651 | 144,905 (4%) | 144,888 (4%) |
| Mahi | 38,052 | 39,566 (4%) | 40,226 (6%) |
| Narmada | 93,494 | 96,660 (3%) | 97,476 (4%) |
| Pennar | 54,243 | 54,905 (1%) | 55,270 (2%) |
| Sabarmati | 30,679 | 31,901 (4%) | 31,642 (3%) |
| Subernarekha | 25,792 | 26,804 (4%) | 26,844 (4%) |
| Tapi | 63,432 | 65,806 (4%) | 66,268 (4%) |
| WFR North & South | 111,629 | 112,591 (1%) | 113,792 (2%) |
|     WFR North | N/A | 58,360 | 58,923 |
|     WFR South | N/A | 54,231 | 54,869 |
| Total | 1,433,516 | 1,467,526 (2%) | 1,473,716 (3%) |

N/A is catchment area information not available

## 3.2 Gauging Station Verification and Landmark Identification

The second geospatial layer in GHI contains metadata on river gauging stations. Station-specific metadata - latitude, longitude, site name and river name, was used to verify the location of each CWC station within Google Maps and/or OSM. CWC's stations are typically named after nearby towns, cities or other landmarks. A reference landmark was identified for each station, whenever possible, and the coordinates of the landmark are included within GHI. Such landmarks provide a definitive reference location which can be re-verified by other users and/or re-positioned if needed. The distance between CWC's station

location and corresponding GHI landmark, and the direction of the landmark relative to the CWC station are also noted.

    An example on the above site verification and landmark identification is presented in Figure 5 for the Neeleeswaram station on Periyar river in the WFR South basin. The CWC location is shown as the red circle. The town of Neeleeswaram is about 4 km to the west of the CWC location and is chosen as the landmark (black square) for this station. The CWC locations do not fall on the pixel-based river network from HydroSHEDS or MERIT due to the approximate nature of these networks. For

feasibility of catchment area delineation and other subsequent analyses, the original station location needs to be relocated on to

**Figure 5.** Example showing GHI site verification and landmark identification for Neeleeswaram station on Periyar river. (a) CWC's station (red circle) is on the right of the graphic. The center of the nearest HydroSHEDS pixel on the GHI river network is the relocated location (blue circle). The landmark is with reference to the CWC location and is on the left (black square). (b) Same as (a) except for the MERIT network. Snippet shows metadata on landmarks and relocated locations included within GHI.

these river networks such that the relocated station is in the middle of a HydroSHEDS (or MERIT) pixel. The relocated station is shown by the blue circle on the river network from HydroSHEDS (Figure 5, panel (a)), and by the green circle on the river network from MERIT (Figure 5, panel (b)).





## 3.3 Catchment Boundary Delineation

The third geospatial layer in GHI contains station-specific catchment boundaries and river networks. Catchment boundaries were derived using the PF-12 watersheds from HydroSHEDS. Using information on upstream watersheds associated with each PF-12 watershed, all the PF-12 watersheds upstream of the relocated station were recursively identified. The polygons corresponding to the most downstream PF-12 watershed and all of identified upstream watersheds were merged (or 'dissolved' in GIS jargon) to create a catchment boundary topographically consistent with HydroSHEDS. At this juncture, the delineated

boundary includes the most downstream PF-12 watershed in its entirety. However, the portion downstream of the relocated station is not needed. Using pixel-specific flow direction information from HydroSHEDS, only the portion contributing to flow at the relocated location was extracted and the remainder discarded. The boundary delineation procedure just described is conceptually similar to standard procedures available within GIS software. The specific procedure used here ensures that the final delineated boundary is consistent with HydroSHEDS' underlying topographic data and is also accurate to the extent

feasible. The station-specific river network was identified by extracting (or 'clipping' in GIS jargon) the HydroSHEDS river network present within the upstream catchment boundary.

A schematic illustrating the above catchment boundary delineation procedure is presented in Figure 6 for the Watrak Dam station in the Sabarmati river basin. The original CWC location (red circle), the relocated GHI station (blue circle) and the GHI river network (blue lines) are overlaid on top of Google Maps (Figure 6, panel (a)). The grey polygons are the HydroSHEDS

PF-12 watersheds upstream of the Watrak Dam station. The portion downstream of the station, not contributing to flow at the station, was discarded and the final catchment boundary was delineated.

The catchment area associated with each station was obtained by estimating the area enclosed by the delineated HydroSHEDS-based catchment boundary. An additional estimate of catchment area was obtained using the MERIT 90 m raster layer on upstream catchment area. CWC's stations were relocated on to the MERIT river network. The catchment area corresponding to

the station was estimated as the raster value at the 90 m pixel containing the relocated location. The HydroSHEDS and MERIT relocated station locations, and the estimated upstream catchment areas are included within the metadata available through GHI.

## 3.4 Quality Control

The quality control process used to assess the reliability of station-wise metadata involves several steps as outlined in Figure 7

and is described in the following text. There are 10 quality checks (QC) and 1 data check (DC), and each station is assigned a 'P' ('P' for pass) or an 'F' ('F' for fail) corresponding to each of these 11 checks. If a station passes a specific check, then it is flagged as 'P' for the particular check. Otherwise, the station is flagged as 'F' for the particular check. A station is placed in Group 3 if it fails to meet any one of the 10 QCs. Otherwise, stations are placed in Group 1 or 2, based on the availability of streamflow data (i.e., based the status of the only DC). Thus, metadata associated with stations in Group 3 should be considered

the least reliable. Stations in Group 1 and Group 2 are equally reliable when it comes to metadata, but Group 1 has reliable daily streamflow data available.

**Figure 6.** Example illustrating GHI's boundary delineation process. (a) CWC's Watrak Dam station (red circle) on the Sabarmati river and GHI relocated location (blue circle). (b) Initial delineated catchment boundary using PF-12 watersheds. (c) Final catchment boundary after discarding the portion downstream of the relocated station.

CWC active monitoring stations (645 stations)

**Quality Check**
P: **Pass**
F: **Fail**

Verify station description (site name and river name(s)) using Google Maps, OpenStreetMap, and Google Search; Identify/add landmarks consistent with station description

Relocate CWC station on to HydroSHEDS Network (500 m); Delineate upstream catchment boundary; Identify river network; Estimate catchment area

Relocate CWC station on to MERIT Hydro Network (90 m); Estimate catchment area

P ← Lat. and Long. available? Lat. and Long. not spurious? — QC1 → F

P ← CWC station present in the correct parent basin? — QC2 → F

P ← CWC station description unambiguously verifiable? — QC3 → F

P ← Can a landmark matching the station description be found? — QC4 → F

P ← GHI landmark within 20 km of CWC station location? — QC5 → F

P ← GHI relocated station location within 5 km of CWC station location? — QC6 → F

P ← CWC catchment area estimate available? — QC7 → F

P ← GHI-derived catchment area within +/- 80% of CWC catchment area? — QC8 → F

P ← GHI-derived catchment area within +/- 10% of MERIT Hydro catchment area? — QC9 → F

P ← HydroSHEDS river network consistent with rivers from online maps? River deltas or other complexities not present? — QC10 → F

Daily streamflow data available from WRIS? (minimum missing records, no extreme outliers) — F → Group 2 (259 stations)

P ↓

Group 1 (213 stations)

Group 3 (173 stations)

**Figure 7.** Overview of GHI's quality control process used in the categorization of gauging stations. CWC's inventory of stations are placed in Groups 1, 2 or 3 based on the outcome ('P' for pass, 'F' for fail) of each quality check.





QC1 is on the availability and reliability of station coordinates - latitude and longitude. Stations with missing or spurious coordinates are placed in Group 3. Latitude and longitude from CWC are presented in the 'DDMMSS' format. The numerical value of minutes and seconds under such a format should span the values from 0 to 60. When values violate such bounds, the coordinates are considered spurious. QC2 ensures that CWC's description of the station is broadly consistent with its coordinates. If the river basin specified by CWC did not match the composite basin associated with the station's coordinates, then it was placed in Group 3.

QC3 ensures that CWC's description of the station is verifiable using either Google Maps or OSM. CWC's description includes the name of the station, the name of the main river (and sometimes a tributary) on which the station is located. Within a general vicinity of 50 km around the station coordinates, a visual search was performed for the namesake river body and a namesake village, town or other landmark (such as a bridge or a highway). The visual search was performed first using Google Maps, and if needed using OSM. An approximate match was also considered acceptable when searching for names of rivers and places. Sometimes, CWC's stations were nowhere near a water body, and sometimes one station's location exactly coincided with the location of another station with a different name. There were also instances when a station was present at a river confluence and it was not evident if the station was intended to be downstream of the confluence or upstream of the confluence (on one of the river branches). One such example is presented in Figure A5 in Appendix A. All such ambiguous situations resulted in the station being placed in Group 3.

QC4 is related to QC3 but is an independent check on CWC's station names. QC4 ensures that a reference landmark matching the station name can be found, regardless of whether a river body is present in the vicinity or not. If a reliable landmark was not found, the station was placed in Group 3. QC5 ensures that the identified reference landmark is not far from the original CWC station. Some of CWC's stations began operation more than 50 years ago. It is possible that population centers (and their names) have experienced changes during this time, and do not always reflect those shown within present-day Google Maps or OSM. A distance of 20 km was selected as a reasonable threshold based on an examination of the typical distances between stations and their landmarks. A majority of the stations meet this criterion and were within 5 km of the corresponding GHI landmark (Figure 8, panel (a)). If a station was more than 20 km away from its landmark, it was placed in Group 3.

During the catchment delineation process, stations were relocated on to the pixel-based river networks. QC6 ensures that the relocated station is in the proximity of the original CWC location. Typically, the relocated stations are only a few pixels away from the original location. However, the relocation distance was much larger on some occasions. Given that some of the larger rivers can have channels spanning multiple HydroSHEDS pixels, a distance of 5 km (approximately 10 HydroSHEDS pixels) was assumed to be a reasonable relocation distance. A majority of the stations meet this criterion and were relocated less than 1 km (Figure 8, panel (b)). If a station was relocated more than 5 km, it was placed in Group 3.

QC7, QC8 and QC9 pertain to availability and reliability of catchment area estimates from CWC. Catchment area was not available for certain CWC stations and QC7 ensures that such stations are placed in Group 3. As discussed in Section 1.1, CWC's catchment area estimates are not reliable, and sometimes could have more than a 50% discrepancy. The discrepancy between GHI's catchment area estimates and those from CWC and MERIT are shown in Figure 8. Based on this chart and an examination of other discrepancies, a discrepancy of 80% between GHI and CWC (GHI relative to CWC) was considered

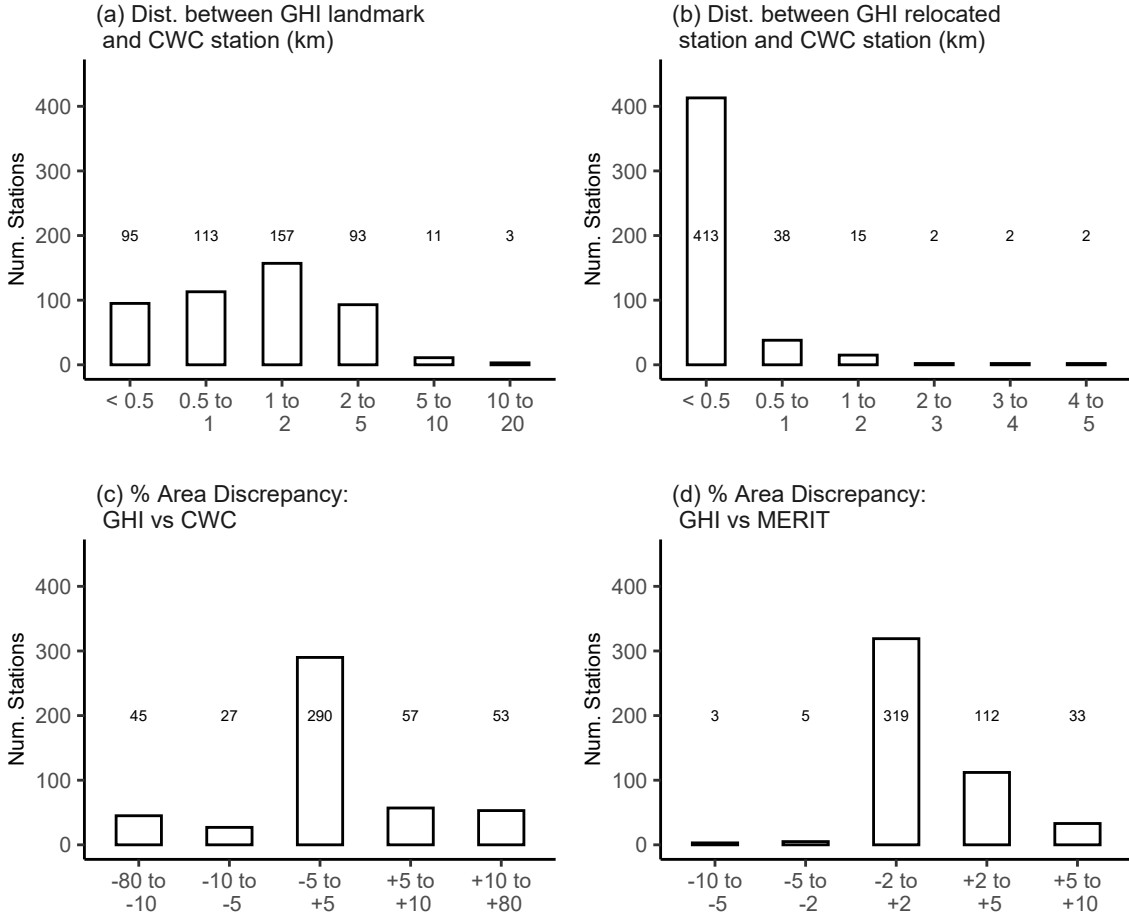

**Figure 8.** Summary of select quality control metrics for stations in Groups 1 and 2: (a) QC5: distance between GHI landmark and the corresponding CWC station; (b) QC6: distance between GHI relocated station and the corresponding CWC station; (c) QC8: discrepancy in estimated catchment area between GHI and CWC; (d) QC9: discrepancy in estimated catchment area between GHI and MERIT; Numbers within each panel indicate the number of stations associated with each bar of the barplot.

an appropriate threshold. Since MERIT-based catchment area estimates are considered equally reliable as those from GHI, a smaller discrepancy of 10% between GHI and MERIT (GHI relative to MERIT) was considered an appropriate threshold. QC8 and QC9 ensure that the catchment area estimates from GHI stay below these thresholds.

QC10 is on the adequacy of the delineated HydroSHEDS-based river network upstream of the gauging station. While the HydroSHEDS river network typically resembles the actual river network based on Google Maps or OSM, it can be grossly inadequate, particularly in the presence of braided river channels, distributaries or other complex flow regimes. The reasoning behind the use of QC10 is to qualitatively assess the adequacy of the delineated network by visually comparing it with actual river flow paths from Google Maps and OSM. A number of stations in the river deltas of Mahanadi and Cauvery basins, and a number of stations on smaller tributaries did not pass this check. One such example is presented in Figure A5 in Appendix A.

The final check is on the adequacy of daily streamflow data. As discussed earlier, for many of CWC's stations in Peninsular India, daily streamflow data is available through WRIS. However, there are often many missing records. It should be noted that a flow value of zero was not considered as missing data in this study. Missing values from WRIS are blanks in the downloaded data spreadsheets. When available, daily data was aggregated to monthly and annual data. Only months with a maximum of 5 missing days and years with a maximum of 60 missing days were used. This would mean that data would be available for at least 80% of the days in a typical month, and 80% of the days in a year. Finally, only those stations with a minimum of 5 years of annual streamflow data were selected. Since water resources assessments and other hydrologic studies often desire sufficiently long time series, a minimum threshold of 5 years was used in this study. However, these 5 years need not be sequential.

The compiled annual time series was checked against statistics presented by CWC-YB for select stations to ensure consistency between the two sets of estimates. A self-consistency check was also performed to ensure there were no spuriously large values. After a visual examination of annual time series of streamflow across all available stations, a value of two times the $95^{th}$ percentile was deemed to be a reasonable threshold. Two stations - Pathagudem in Godavari basin and Sadalga in Krishna basin, had annual values that exceeded the above threshold for WY 2017. Both annual and monthly data for WY 2017 for these two stations was discarded.

The number of stations falling within each group are tabulated in Table 7. Out of the 645 stations analyzed here, 213 are in Group 1, 259 are in Group 2 and the remaining 173 are in Group 3. Thus 472 of the 645, or about 73% of the stations are in Group 1 and 2. Except for the Pennar basin, all river basins have at least one station falling in Group 3. All of the basins have at least a few stations in Group 1. Figure 9 shows the spatial distribution of stations across groups. Panel (d) of Figure 9 shows a cluster of stations in the Mahanadi and Cauvery deltas. These stations are associated with river networks which are too complex for GHI to delineate. Stations in the WFR South basin correspond to catchment areas that are typically much smaller than those in other basins. This is due to the Western Ghats (or mountains) present along the coast which result in many small coastal watersheds along the Western Coast of India. Several such catchments of the WFR South basin have catchment area discrepancies larger than the thresholds defined in the quality control process (QC8 and QC9 in Figure 7). Such stations typically have catchment areas of a few hundred square kilometers or less. As discussed in Section 6, the 500 m resolution of HydroSHEDS data is probably too coarse to adequately resolve the boundaries and river networks of such catchments.



**Table 7.** Distribution of gauging stations by GHI group within each composite basin.

| Basin | Group 1 | Group 2 | Group 3 | Total |
|---|---|---|---|---|
| Brahmani-Baitarani | 7 | 8 | 9 | 24 |
| Cauvery | 20 | 9 | 25 | 54 |
| EFR North | 5 | 10 | 5 | 20 |
| EFR South | 12 | 13 | 12 | 37 |
| Godavari | 40 | 70 | 30 | 140 |
| Krishna | 42 | 22 | 8 | 72 |
| Mahanadi | 19 | 19 | 17 | 55 |
| Mahi | 6 | 10 | 3 | 19 |
| Narmada | 18 | 34 | 19 | 71 |
| Pennar | 6 | 6 | 0 | 12 |
| Sabarmati | 2 | 9 | 2 | 13 |
| Subernarekha | 4 | 7 | 4 | 15 |
| Tapi | 4 | 21 | 15 | 40 |
| WFR North | 4 | 13 | 5 | 22 |
| WFR South | 24 | 8 | 19 | 51 |
| All Basins | 213 | 259 | 173 | 645 |

## 3.5 Time Series Data Compilation

Stations passing the above described quality control measures fall in Group 1 or 2, and are much more reliable than those in Group 3. Only these stations in Groups 1 and 2 are used in subsequent analyses. Station-wise monthly and annual time series data of select hydrometeorological variables are compiled. In order to facilitate comparison across variables, they are expressed in the same units of volume. Various reports from CWC (e.g., CWC-19 and CWC-YB) present summary statistics on streamflow (and other hydrometeorological variables) in units of $\mathrm{MCM}$ (or million $\mathrm{m}^3$). Hence, this unit was chosen to facilitate cross checking. Thus, precipitation, modeled ET and runoff, and observed streamflow are expressed in units of $\mathrm{MCM/month}$ or $\mathrm{MCM/year}$ within GHI. For the purposes of discussion and graphical display, sometimes the unit of billion cubic meters (BCM, equivalent to $\mathrm{km}^3$) was also used. Gridded data on precipitation, modeled ET and runoff are available in units of depth per unit area per month (e.g., $\mathrm{mm/month}$), and these were converted to cumulative monthly and annual volumes by accounting for grid-specific area.

Modeled grid-level runoff is often aggregated to the catchment and compared with observed streamflow in hydrologic analyses. Such aggregation accounts for travel time from individual grids to the catchment outlet using a channel flow routing procedure. For this compilation, the monthly and annual runoff is assumed to be unaffected by channel routing. Modeled




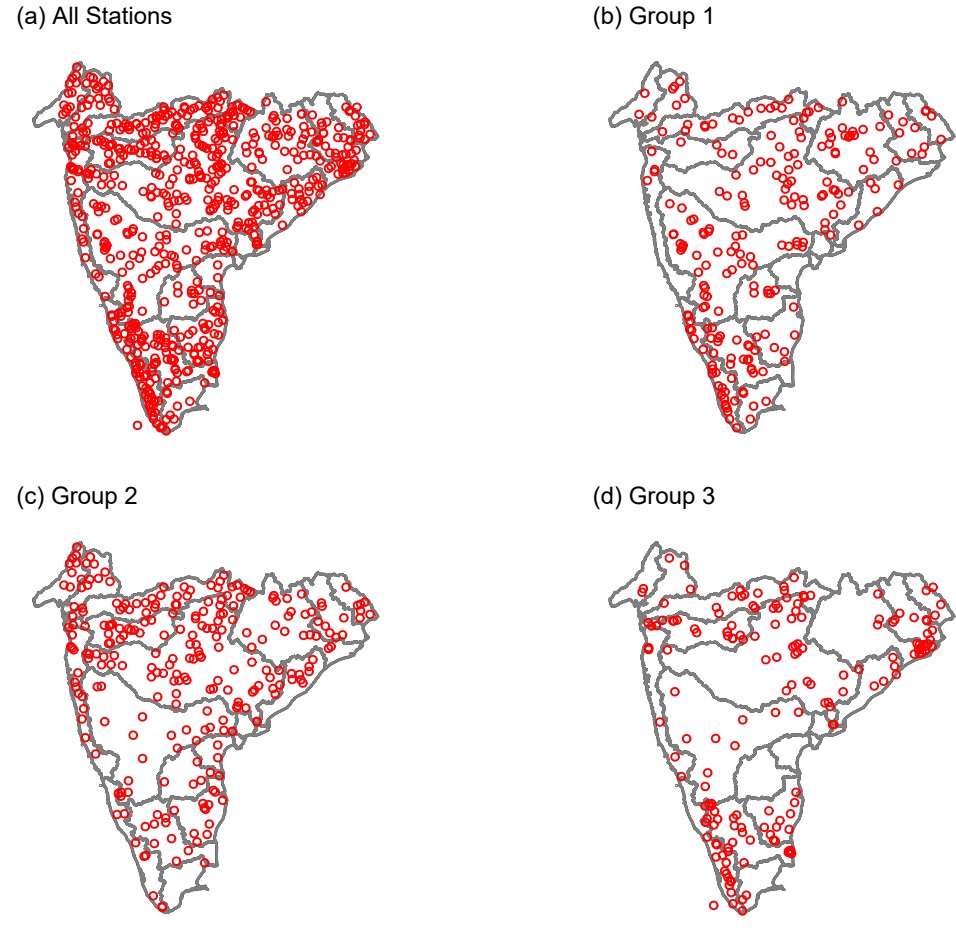

**Figure 9.** Stations falling with each GHI group (a) All Stations; (b) Group 1; (c) Group 2 and (d) Group 3.

streamflow was simply estimated as the grid area-weighted runoff over the entire catchment. For the downstream locations of large river basins such an assumption may not be appropriate.

The time span of this data compilation is WY 1950 to 2020 (71 water years), whenever data is available. The Summer Monsoon typically starts in the month of June and marks a big seasonal shift in the climate of the study region. Consistent
with CWC-19 and other CWC publications, a water year (WY) is defined as the period starting from June and ending in May of the following calendar year. For example, WY 2020 spans the period June 1 2020 through May 31 2021. Precipitation data from IMD, and all data from ERA span the 71 WYs, while GLEAM data spans WY 1980 to 2020 (41 WYs). Availability of streamflow data from WRIS is station dependent, and no station spans the entirety of the above 71 WYs. The compiled streamflow observations for stations in Group 1 span WY 1965 to 2017, with individual station record lengths varying from 6





WYs to 53 WYs (median length of 34 WYs). There are 175 stations within GHI which have at least 20 WYs of streamflow data.

## 4 Final Product

GHI is publicly available (Goteti (2023)) and includes metadata on stations, GIS data on catchment boundaries and river networks, and station-wise summary graphics. Additional information on files included within GHI is in Section 7.

While users of this dataset can filter down to the specific station or river basin of interest, it is useful to have readily available summary graphics which provide an overview of catchment-scale water balance metrics. Such summary graphics not only provide a visual check on the data created for this station, but could also be useful tools for water managers and other stakeholders. Two sets of summary graphics are included within GHI - a station-wise annual time series chart and a monthly time series chart. An example for the Mancherial station in the Godavari basin is shown in Figures 10 and 11. Figure 10

includes a map of the station location and also a map of the upstream catchment area and the river network. The charts show the time series of catchment-averaged hydrometeorological data available through GHI.

From Figure 10 it is evident that annual precipitation from IMD and ERA are generally consistent with each other. Estimated ET from ERA and GLEAM are also temporally consistent with each other, but consistently differ in magnitude for this station. While the specific causes of discrepancies between these two datasets are unknown, the reader is referred to Muñoz-Sabater

et al. (2021) and the references therein for further discussion. Annual runoff from ERA tends to be lower than observed runoff prior to 1990, but tends to be higher after 1990. While the exact cause of this is unknown, it is speculated that flow regulation by dams and reservoirs could be one of the reasons. ERA does not account for flow regulation by dams and reservoirs and is not expected to match the observed flow. Overall, ERA reasonably captures annual-scale observed precipitation (from IMD) and observed runoff (streamflow from WRIS).

The monthly time series of hydrometeorological variables for the Mancherial station is shown for the most recent ten year period in Figure 11. The monthly time series for each year in the chart begins in June of the current year and ends in May of the following calendar year, consistent with the definition of water year used throughout this study. Similar to Figure 10, precipitation from ERA and IMD are highly consistent with each other. ET from ERA is higher than that from GLEAM, particularly at the beginning of the water year (summer and fall months). As mentioned earlier, ERA does not account for

flow regulation and could be one of the reasons why ERA's runoff is higher than observed flow. The monthly time series of precipitation, ET and runoff reflect the seasonality imposed by the southwest monsoons - wet season from June to September, followed by a dry season.

Similar graphics are available for the rest of the stations in Group 1 and Group 2. While users can visually examine graphics on individual stations, it is not feasible to assess the overall adequacy of the compiled hydrometeorological through visual

examination alone. A preliminary analysis was performed in Section 5 using the compiled hydrometeorological data. The aim of this analysis was to check for the presence (or absence) of patterns expected based on the hydrology and climate of the study



**Figure 10.** One page summary of final GHI output for Mancherial station in the Godavari basin. Top panel: (left) shows the location of the Mancherial catchment within the Godavari basin, and (right) Mancherial catchment and river network. Bottom panels: time series of precipitation, evapotranspiration and runoff by water year (WY).

**Figure 11.** Same as 10 except showing monthly time series of compiled hydrometeorological data for WY 2011-2020.

domain. Such an analysis would also reveal any spurious patterns within the compiled time series data and help understand the consistency between different datasets.

# 5 Preliminary Analysis using GHI

The analysis presented here focuses only on catchment-averaged annual-scale metrics for the sake of simplicity. Metrics examined include correlation between precipitation from ERA and IMD, correlation between ET from ERA and GLEAM, ratio of observed runoff and precipitation. Only stations in Groups 1 and 2 (472 stations) were considered when analyzing precipitation and ET, while only stations in Group 1 (213 stations) were considered when analyzing observed runoff.

The linear correlation coefficient (Pearson correlation using pairwise complete data) between IMD's annual precipitation and
ERA's annual precipitation is shown in Figure 12. For a majority of the stations (331 out of 472) the correlation is between 0.50

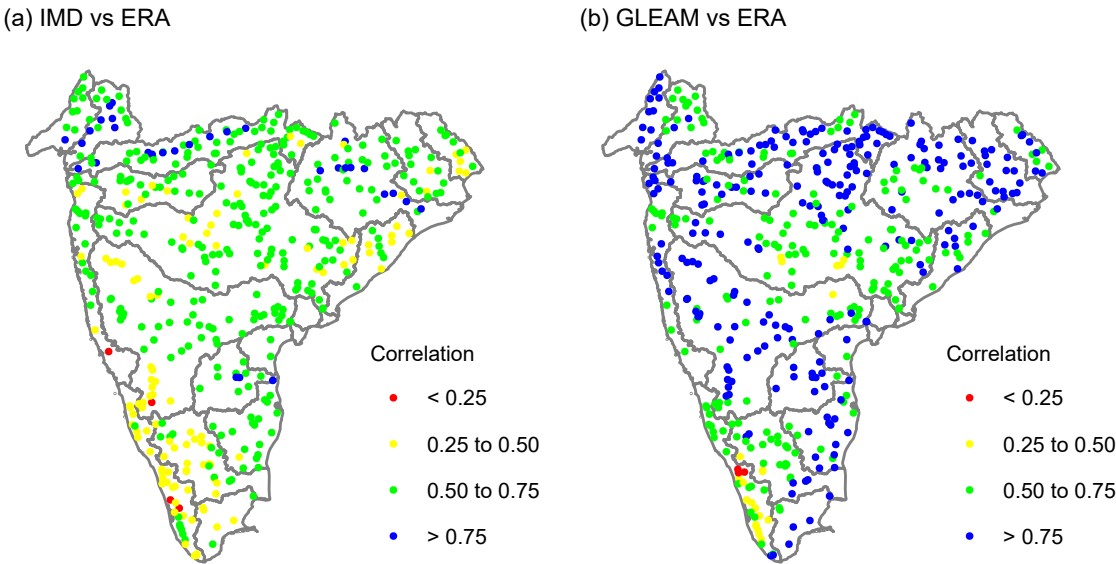

**Figure 12.** (a) Correlation between annual catchment-averaged precipitation from IMD and ERA; (b) correlation between annual catchment-averaged ET from GLEAM and ERA.

and 0.75. Correlation is greater than 0.75 for a small number of stations (31 out of 472). Correlation is lower in the southwestern portion of the study domain where hilly terrain is typical - this includes Southern Krishna basin, Western Cauvery basin, parts of WFR North basin, and almost the entirety of WFR South basin. The general consistency between IMD and ERA precipitation was also reported by Mahto and Mishra (2019) and Goteti (2022).

The correlation between annual ET from GLEAM and ERA is also shown in Figure 12. For a majority of the stations (250 out of 472) the correlation is greater than 0.75. Similar to precipitation, the correlation is higher in other parts of the study domain, compared to the southwestern portion of the study domain. The consistent high correlation between GLEAM and ERA estimated ET was also noted by Goteti (2022).

Figure 13 shows median ratio of observed runoff and precipitation ($R/P$) from IMD and ERA. For each year, $R/P$ was 475     computed using available data, and the median value was estimated and plotted in Figure 13. In general, for a majority of the stations (125 out of 213 for IMD, and 136 out of 213 for ERA), median $R/P$ is less than 0.33. This is expected given the semiarid climate of a large part of the study domain. In such climates, a large portion of the annual precipitation would go towards satisfying the evaporative demand, resulting in low $R/P$ ratios. For the southwestern portion of the study domain, where hilly terrain and wet climate are prevalent, $R/P$ ratios are expected to be higher. This was the case for several stations. 480     However, median $R/P$ is greater than 1.0 for certain stations, regardless of whether the precipitation was from IMD or ERA. Such stations are present mostly in the hilly regions along the Western Coast of India. A $R/P$ value greater than 1.0 implies

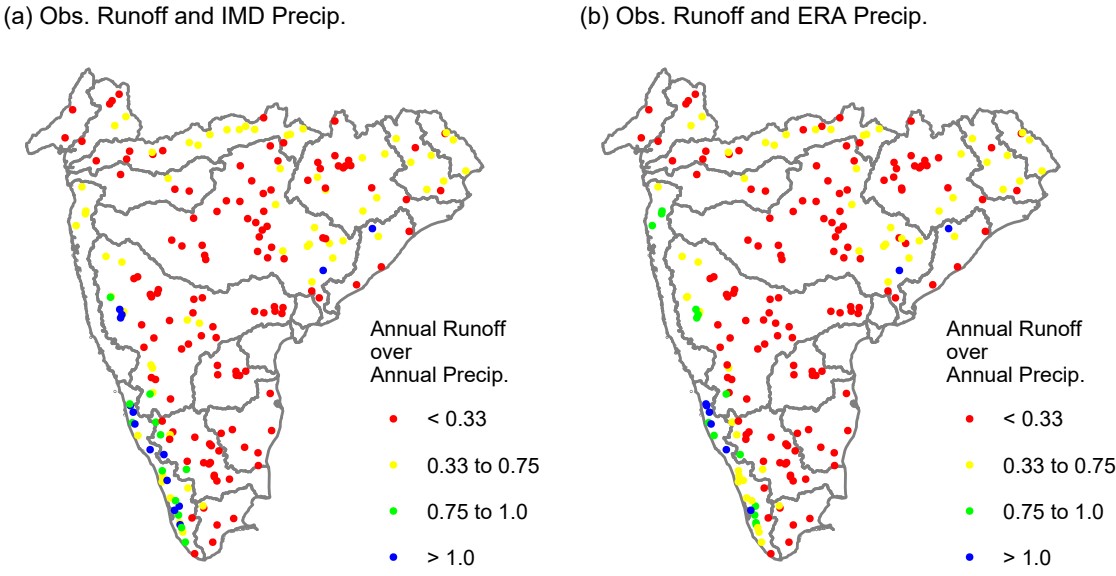

**Figure 13.** (a) Median ratio of observed annual runoff and IMD annual catchment-averaged precipitation; (b) median ratio of observed annual runoff and ERA annual catchment-averaged precipitation;

that total annual runoff from the catchment exceeds the total annual precipitation. Such a finding may seem 'absurd' at a first glance and warrants further discussion.

From basic hydrologic balance, in the absence of any human intervention, the annual volume of runoff from a river basin is expected to be less than the annual volume of precipitation. Hence, values of $R/P$ greater than 1.0 suggest either substantial human intervention to the water cycle (e.g., carryover reservoir storage) or errors associated with data. Such errors could be with catchment area delineation, erroneous data compilation, or erroneous underlying runoff or precipitation data. Catchment area discrepancy between GHI and CWC-21 for stations where $R/P$ is greater than 1.0, is within 5% for most of the stations. Moreover, precipitation and runoff data compiled from this study is consistent with independent compilations from other studies (CWC-YB and CWC-19). Hence, data compilation errors can be ruled out.

It is speculated that the 'absurdity' of $R/P$ being greater than 1.0 is either due to carryover reservoir flow or due to errors with the underlying streamflow or precipitation data. It is not known whether there are any gross measurement inaccuracies associated with streamflow data. Other studies have indicated the complexities of capturing precipitation in this region where orographic effects play a major role in creating intense precipitation events (e.g., Rana et al. (2015) and Thakur et al. (2019)). The estimated catchment areas of these stations with spurious $R/P$ ratios are typically less than 1,000 $\mathrm{km}^2$. Since IMD's individual grids are about 25 km by 25 km or 625 $\mathrm{km}^2$ in size, IMD's data may not be suited to capture the necessary spatial variability in precipitation. However, ERA's precipitation suffers from a similar issue despite being at a higher resolution (grid size of about 100 $\mathrm{km}^2$). It should also be noted that this is not the first study to encounter spurious $R/P$ values in this region.





CWC-19 tabulated annual runoff values greater than annual precipitation for some of the catchments of the WFR South river basin (CWC-19 (2019), see Appendix R, tables R-1 to R-10), but did not delve into the underlying causes. A further discussion on this topic is beyond the scope of this paper and will be addressed in the future.

## 6    Limitations and Potential Improvements

GHI's adoption of 15 arc-second (500 m) HydroSHEDS data as the underlying template imposes a limit on the spatial accuracy of delineated catchment boundaries and river networks. For larger river basins, this resolution is probably adequate

for most practical purposes. However, for smaller river basins, such as those less than $100 \mathrm{~km}^2$ (equivalent to about 400 HydroSHEDS pixels), topographic data based on higher resolution might be more appropriate. A new version of HydroSHEDS (v2, https://www.hydrosheds.org/) based on 12 m topographic data is scheduled to be released in 2023. It is expected that this new version would improve the spatial accuracy of the entire suite of HydroSHEDS products. GHI's catchment boundaries and river networks could be updated with this latest dataset. While significant effort was made to accurately identify the catchment

boundaries corresponding to each station, there were instances where the delineated boundary included a few 500 m pixels not contributing to flow at the station. The effect of such pixels on delineated catchment boundaries and area estimates is minimal, but they need to be discarded to make the boundaries more accurate. In future revisions, this issue will be addressed.

   Some of the quality checks used in the development of GHI use subjective thresholds, and were devised to separate the more reliable data from the less reliable. One could end up with a different number of stations within each group if those

subjective thresholds were changed. Out of the 645 stations analyzed here, 173 stations were placed in Group 3 because sufficient reliable metadata could not be compiled for many of these stations. It is hoped that CWC would corroborate metadata on these stations so that they could eventually be moved into Group 1 or 2. An obvious next step is to extend the domain of GHI from Peninsular India to the whole of India. However, streamflow data for the rest of India is not publicly available. Such data is critical for hydrologic assessments, climate change studies and other modeling analyses. It is also hoped that this study would

encourage CWC and other custodians of streamflow data to make such data publicly available. Another issue with extending the domain of GHI to the whole of India is the issue of drainage boundaries crossing international boundaries and the prevailing uncertainty about these drainage boundaries. An analysis on Ganga's basin boundaries delineated using HydroSHEDS data and discrepancies with those available from CWC and WRIS is currently under review (submitted to Current Science, September 22, 2022).

Missing daily streamflow values were filled in using the average of available daily data. Such a procedure will not adequately capture the rising and falling limbs of the daily hydrograph. Future revisions could use established methods such as time series interpolation of Fritsch and Carlson (1980), which is readily available within many data analysis software. Currently, hydrometeorological data within GHI includes precipitation, modeled ET and runoff, and observed streamflow. Additional variables such as catchment-averaged soil moisture could be useful in water balance studies. Agriculture is the biggest land

use in India (NRSC (2007)) and is responsible for about 70% of water abstractions (CWC-19). Inclusion of annual maps of





cropland and irrigated area would be useful in the estimation of consumptive water use within water resources assessments, such as those by CWC-19 and Goteti (2022).

While event-scale precipitation over the Western Coast of India has been analyzed before, such as the extreme precipitation resulting in the floods of 2018 (e.g., Hunt and Menon (2020)), this is probably the first study to report potential annual-scale
under representation of precipitation in this region. This issue needs to be investigated further. High-resolution datasets on precipitation, such as those from IMDAA (Rani et al. (2021)) and CHIRPS (Funk et al. (2014)) could be helpful in addressing such issues. Human management of water could have a substantial effect on streamflow and the hydrologic cycle. The inclusion of data on dams and reservoirs (both metadata and live storage) would be helpful in quantifying the effects of flow regulation. WRIS-OL provides live storage information for select reservoirs and this data could be included within GHI in future revisions.

**7  Data availability**

GHI is publicly available (Goteti (2023)) at https://doi.org/10.5281/zenodo.7563599 and includes the following files: (1) a 'Readme.txt' file which outlines the available files, the format of these files, and the data fields within these files; (2) plain text files on station metadata (station locations from CWC, relocated locations, landmarks, catchment areas and other attributes) and hydrometeorological data time series (monthly and annual files); (3) shapefiles (GIS) on composite river basin boundaries,
catchment boundaries and catchment-specific river networks for stations present in Group 1 and Group 2; and (4) PDF files showing station-wise summary maps and time series (monthly and annual files).

**8  Conclusions**

A fundamental building block of hydrologic analyses is GIS data on river gauging stations, their upstream catchment area boundaries and river networks. The limited available information for India's river basins suffers from drawbacks such as
ambiguous station locations, inconsistent or erroneous catchment area estimates, among others. The goal of this study is to highlight the limitations of existing data and to build a publicly available hydrographic dataset using state-of-the-art global resources. The dataset developed by this study, GHI, categorizes available information from India's water agencies based on its consistency with global data sources. Existing metadata is supplemented with additional information where needed. The quality control aspect of GHI includes: verifying station description using online maps, checking for visual consistency of
delineated river network with online maps, comparing GHI-estimated catchment areas with those from CWC and MERIT, and checking streamflow data for missing records and extreme outliers.

The current version of GHI is limited to 645 stations in 15 river basins of Peninsular India. Out of these 645 stations, 472 were deemed reliable for subsequent analyses and 173 were not. While the geospatial information within GHI includes GIS data on gauge locations, catchment boundaries and river networks, the time series information within GHI includes precipitation,
ET and runoff at monthly and annual timescales for each of the above 472 stations. A preliminary analysis using GHI's time series data suggests that while the compiled data is reasonable over most of the study domain, spurious runoff-precipitation



ratios were observed in the hilly coastal regions of Western India. This issue needs to be investigated further. Building a robust hydrographic and hydrometeorological dataset is beyond the scope of one individual. Until such datasets become available, GHI is intended to serve as a building block and a reliable reference for hydrologic analyses on India's river basins.

*Author contributions.*  All analyses were performed by the sole author of this paper.

*Competing interests.*  No competing interests are present.

*Acknowledgements.*  A number of publicly available datasets were used in this study and were cited wherever applicable. Software used in this study includes the R statistical computing and graphics software for data analysis (https://www.r-project.org/) and QGIS for GIS analysis
(https://qgis.org/en/site/). Political boundaries for India were obtained from the Survey of India (https://surveyofindia.gov.in/pages/outline-maps-of-india) and used for illustration only.



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



## Appendix A: Additional Graphics on GHI's Data Sources

| State : Tamilnadu | | | | | | River Basin: Cauvery | | | | |
|---|---|---|---|---|---|---|---|---|---|---|
| Site Name | District | HO/FF/ HOIW | Catchment Area SqKm | River Name/ Tributory/ SubTributory | Type | Date of Start | Latitude | Longitude (a) CWC-21 | | Site Code |
| Biligundulu | Krishnagiri | HO | 36682 | Cauvery | GDSQ | 01/07/1971 30/08/1971 01/06/1972 01/06/1978 | 12°10'56" | 77°43'26" | | CW1CAM000 |
| Elunuthimangalam | Erode | HO | 3386 | Cauvery/Noyyal | GDSQ | 27/11/1997 07/08/1998 01/03/2013 03/10/2000 | 11°01'54" | 77°53'15" | | CW1CAM000 |

| River Point Level & Flow Report (ALL AGENCIES) From 20070101 to 20071231 | | | (b) WRIS-OL | | | | |
|---|---|---|---|---|---|---|---|
| | | | 01 Jan 2007 | | 02 Jan 2007 | | |
| River Point Name | Latitude | Longitude | Level (m) | Flow (cumecs) | Level (m) | Flow (cu |
| Sagjuri | 21.05027778 | 84.05583333 | | | | |
| Salebhata | 20.98333333 | 83.53944444 | 130.4 | 1.44 | 130.4 | |
| Samal | 21.08388889 | 85.13055556 | | | | |
| Saradaput | 18.6 | 82.13333333 | 227.34 | 73.93 | 227.27 | |
| Saradaput | 18.62027778 | 82.11666667 | | | | |
| Sitalpur | -14.028580744741944 | 49.285250431147475 | | | | |
| Sarada | 10.7590555C | 84.45222222 | | | | |

**Figure A1.** Snippet showing the metadata information available from CWC and WRIS. (a) Data from CWC-21; (b) Raw downloaded streamflow data from WRIS-OL for the year 2007. Daily data from WRIS-OL is sometimes available as gauge level and/or discharge. Note the discrepancy in latitude and longitude for the Saradaput station. Also, note the spurious latitude and longitude for the Sitalpur station (as such corresponds to a location in the Southern Hemisphere).

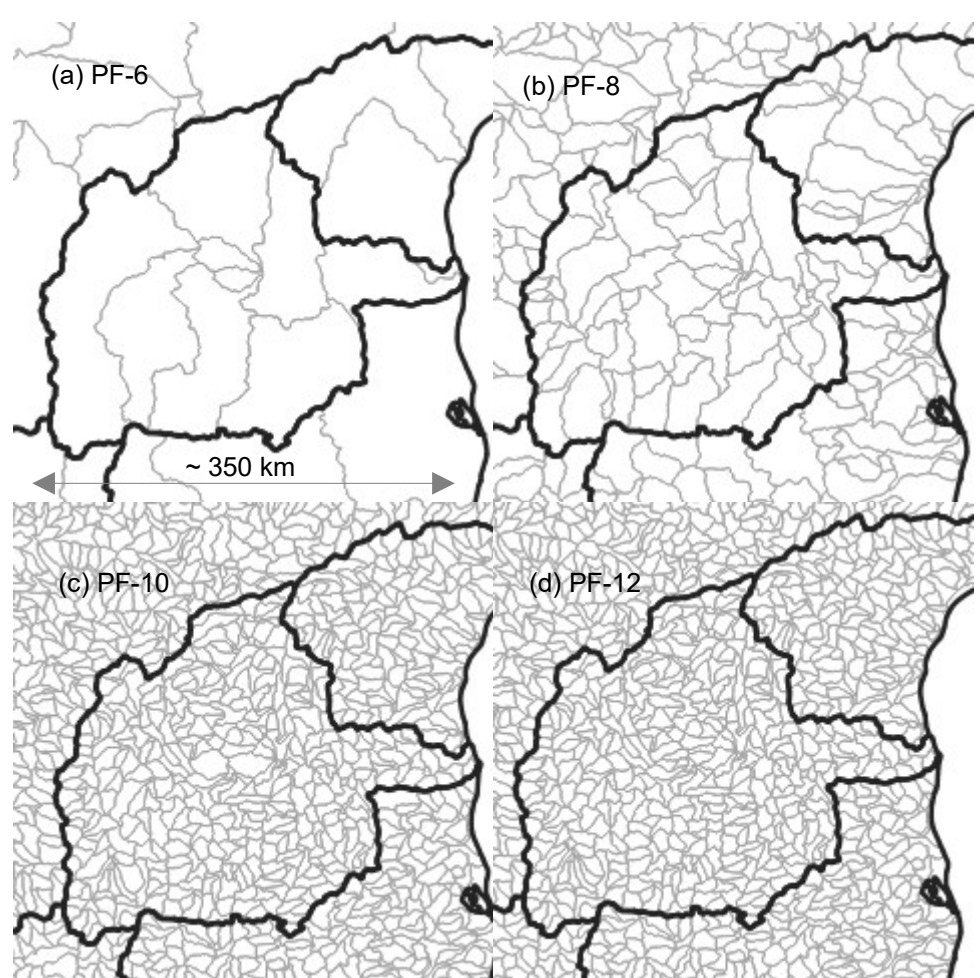

**Figure A2.** HydroSHEDS watersheds (grey lines) across Pennar river basin (black line) at different Pfafstetter (PF) levels: PF-6, PF-8, PF-10 and PF-12.

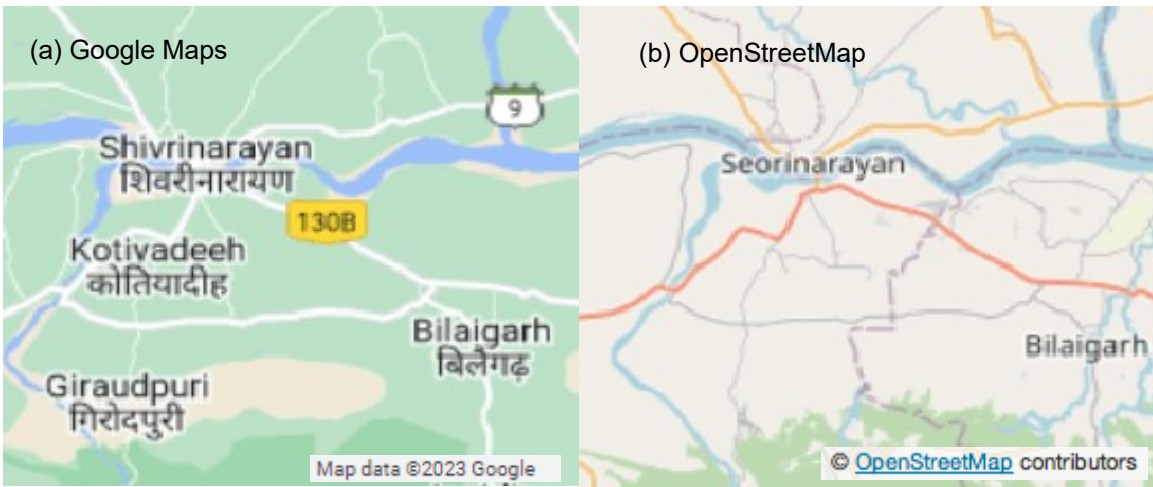

**Figure A3.** Example showing the typical information available from (a) Google Maps and (b) OSM for a selected region in the Mahanadi river basin. Google Maps typically has more town and city names, while OSM has more natural features such as river and water bodies.



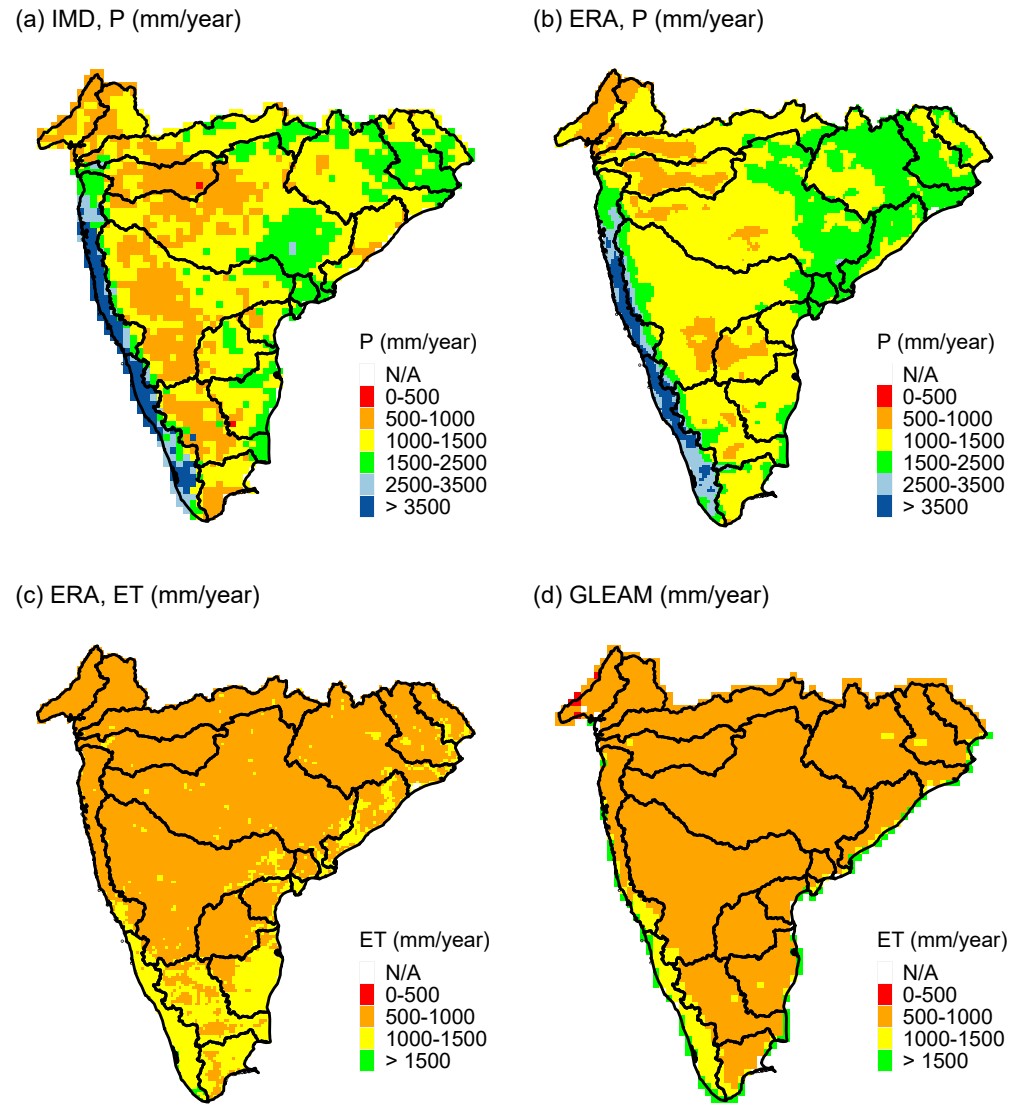

**Figure A4.** Total annual precipitation (mm/year) for WY 2020 (June 2020 through May 2021) from (a) IMD and (b) ERA5. Total ET (mm/year) for WY 2020 from (c) ERA5 and (d) GLEAM. 'N/A' in the legend indicates missing or unavailable data. IMD and GLEAM are shown on the native 0.25° grid while ERA data is shown on the native 0.10° grid.



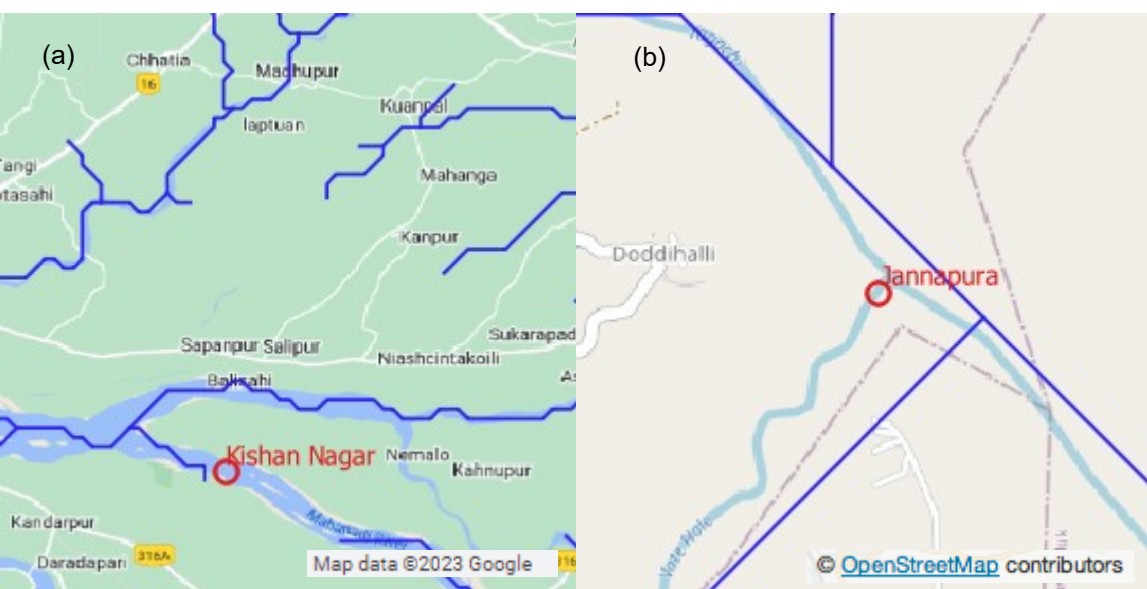

**Figure A5.** (a) Kishan Nagar station (red circle) is on one of the distributaries within the Delta of the Mahanadi River. HydroSHEDS river network (blue line) cannot capture such complex features (background from Google Maps). (b) Jannapura station in the Cauvery basin (red circle) could not be relocated on to the HydroSHEDS network (blue line) because it was unclear whether the station is intended to be upstream or downstream of the confluence (background from OSM).