# Peer review of "Geospatial dataset for hydrologic analyses in India (GHI): A quality controlled dataset on river gauges, catchment boundaries and hydrometeorological time series"

_Earth System Science Data, 2023_

## Referee Comment (RC2)

Review of the manuscript '***Geospatial dataset for Hydrologic analyses in India (GHI): A quality controlled dataset on river gauges, catchment boundaries and hydrometeorological time series***' by *Gopi Goteti* submitted to *Earth System Science Data*.

Recommendation: ACCEPT after minor corrections

Focus: quality control of hydrological data which are categorized using existing metadata over India.

Title: correct (don't capitalise) "Geospatial dataset for hydrologic analyses in India (GHI): A quality controlled dataset on river gauges, catchment boundaries and hydrometeorological time series"

Abstract is well written and clearly describes the undertaken study.

Figures The following Figures have bad quality and should be replaced with high resolution images:

1. A5. (a) Kishan Nagar station<...>; (b) Jannapura station;

2. Figure A3.

3. Figure 6. Example illustrating GHI's boundary delineation process;

4. Figure 5. Example showing GHI site verification and landmark identification;

5. Figure 3. (a) Erroneous station location from CWC

Other Figures are acceptable.

Relevance: The presented study is the original primary research within scope of the journal. The manuscript meets general criteria of the significance in hydrological monitoring and data quality control. The study is relevant to the journal topic as corresponding to the major domain and research disciplines.

Data used in this study are described: the author collected data from 472 stations, catchment-specific annual and monthly time series spanning long-time period (1950-2020). he compiled and processed the data using observed precipitation from IMD, observed streamflow from WRIS, estimated precipitation, evapotranspiration (ET) and streamflow from ERA5-Land, and ET from GLEAM. Data sources are well explained. The data components are well summarised in Figure 4 and Table 6 (catchment areas).

Introduction presents a background, defines research goals and provides a clear statement of research problem. The Introduction well describes the research. Introduction and background show context of the article. Literature is well referenced and relevant.

Study area: India.

Research questions and goal are identified: to develop a new dataset, the 'Geospatial dataset for Hydrologic analyses in India' (GHI) which uses HydroSHEDS data as the underlying template. has both. Objectives are relevant to the study aim: to update geospatial and time series information using GHI.

Literature regarding the relevant topics is reviewed, formatted according to the journal rules and appropriately referenced. Major sources include published papers on topics of hydrology, soil and environmental studies, land cover products for hydro-climate modeling, water-related studies and hydrological observations in India, climate-catchment-soil control on hydrological droughts, etc. The literature references are relevant and well cited.

Research gaps and weakness in former works are described: for India's river basins, availability of data on streamflow and metadata on gauging stations, GIS data on station locations, their upstream catchment boundaries and river flow networks is limited; when available, such data are not in an analysis-ready format and can have substantial errors. The existing gaps are identified: studies often use information from India's water agencies without checking for its validity. The contribution of this work filling this gap is explained. It concerns updated hydrological information in India using HydroSHEDS data.

Methods: Methods described with sufficient detail and information. The author performed the quality control process, using CWC's stations in Peninsular India and categorized the data into three groups: 1) reliable metadata, adequate daily streamflow data; 2) reliable metadata, inadequate or no daily streamflow data; 3) missing or unreliable metadata. The workflow is well structured and clearly described with sufficient

information to reproduce the approach. The author evaluated river basins of Peninsular India with publicly available daily streamflow data.

Motivation is explained: this study contributes to fill in the gaps through revealing the limitations of existing hydrological datasets. Thus, he supported the community-led effort towards building the needed datasets for hydrological monitoring in India.

Results are reported: The authors assessed the data from the streamflows and found that the compiled data appears reasonable over most of the study domain, spurious runoff-precipitation ratios were observed in the hilly coastal regions of Western India. The author compiled time series data and presented the results in section 3.5. Quality control metrics for stations is well illustrated in Figure 8 (section 3.4 Quality Control). The overview of the GHI's quality control process used in the categorization of gauging stations is well demonstrated in Figure 7 as a logical scheme. Time series of precipitation, evapotranspiration and runoff by water year and the location of the selected catchment areas with river network are presented. The Results are presented with clarity and include description, graphs, tables, and maps. The results are relevant to the initial research goals and objectives and highlights major achievements of this study.

Discussion interpreted the major outcomes of this study. The advantages of the obtained results are described and compared with other studies on hydrological monitoring in India. The Discussion described the issues of methodology and results.

Conclusion Conclusions are well stated, linked to original research question, limited to supporting results and summarized the study with interpretation of facts: control hydrological data over India to ensure robust and reliable hydrographic and hydrometeorological datasets. The conclusions are appropriately stated and connected to the original questions.

Structure: The article is well organized with structured sections. The structure of the manuscript conforms to the journal standards and discipline norm. It has the following standard sections: Introduction, Methodology, Results, Discussion, Conclusion, References. The numeration of the sections is correct and consecutive.

English language: fine. Clear, unambiguous, professional English language used throughout.

Logic: The clarity of the text logic and organization of the paper is sufficient. It demonstrates the consistent interpretation of the results with detailed explanations and comments. A comparison of the results with those in previous studies is presented.

Actuality, novelty and importance of the research is clear: the importance of streamflow gauging stations consists in tracking the pulse of rivers and also acting as common reference points for hydrologic and other environmental analyses. Therefore, updated information from streamflow gauging stations is essential for environmental and hydrological modelling.

Academic contribution: Rigorous investigation performed in hydrological analysis in Indian to a high technical and professional standard. The paper increases the knowledge in hydrological data analysis by building a new dataset using existing metadata (from CWC and WRIS) and checking it against publicly available information from global data sources (e.g., WWF, MERIT and Copernicus), and online maps (e.g., Google Maps).

Recommendation: This manuscript can be ==ACCEPTED after minor corrections== based on the detailed report above.

With kind regards,

- Anonymous Reviewer.

04.07.2023.

---

## Author Comment (AC1)

**Response to Reviewer #2's comments:**

[only select comments from Reviewer #2 involving edits to the manuscript are shown below]

The author would like to thank the reviewer for the positive feedback and the suggestions made to improve the readability of the manuscript. The author agrees with all of the comments, and revisions were made accordingly. Please see below the specific responses to the individual comments.

Title: correct (don't capitalise) "Geospatial dataset for hydrologic analyses in India (GHI): A quality controlled dataset on river gauges, catchment boundaries and hydrometeorological time series"

The title has been revised accordingly.

Figures The following Figures have bad quality and should be replaced with high resolution images:

1. A5. (a) Kishan Nagar station<...>; (b) Jannapura station;

2. Figure A3.

3. Figure 6. Example illustrating GHI's boundary delineation process;

4. Figure 5. Example showing GHI site verification and landmark identification;

5. Figure 3. (a) Erroneous station location from CWC

Other Figures are acceptable.

In the original manuscript screenshots from Google Maps and OpenStreetMap were used within several figures to provide spatial reference. The displayed screenshots turned out to be blurry in print. In the revised manuscript, such screenshots were replaced with high resolution original graphics (original tiles from the above map vendors). Figures 3, 5, 6, and figures A3 and A5 in the appendix were revised and the author made sure that the printed version of these figures are not blurry.

---

## Author Comment (AC2)

**Response to Reviewer #1's comments:**

This paper aims to enhance both the quantity and quality of the streamflow data and metadata on gauging stations in India. The author integrated multiple datasets to update the information of gauging stations and compile historical streamflow data through a rigorous quality control process. The information and datasets provided in this study will prove valuable to hydrologic communities, while the methodologies presented can be adopted by both the administrative organizations and research communities to homogenize the hydrological datasets in India. I found this paper is generally well written. I recommend this paper to be accepted after few minor comments below are addressed.

The author would like to thank the reviewer for the positive feedback and the suggestions made to improve the readability of the manuscript. The author agrees with all of the comments, and revisions were made accordingly. Please see below the specific responses to the individual comments.

> 1.The GHI dataset is compiled from multiple datasets. It will be good to add a flowchart showing the steps of how different datasets are processed and the relationships between GHI and these datasets. Figure 4 and Table 5 serve such purpose but the usage and relations of different datasets are not clear.

In order to address this comment, Figure 4 was revised to include a chart illustrating the input datasets, the data preparation methods, the specific quality control measures associated with each method, and the final outputs from this study. The beginning of Section 3 ("Methodology") was revised accordingly.

> 2.The base maps in figures are indistinct (e.g., figures 3 and 5). It seems that the author used screenshots of Google Maps and enlarged the images without properly dealing with the resolution. The letters of street names and landmark names are blurry. Please make sure to use high-resolution images for base map when making these figures.

In the original manuscript screenshots from Google Maps and OpenStreetMap were used within several figures to provide spatial reference. The displayed screenshots turned out to be blurry in print. In the revised manuscript, such screenshots were replaced with high resolution original graphics (original tiles from the above map vendors). Figures 3, 5, 6, and figures A3 and A5 in the appendix were revised and the author made sure that the printed version of these figures are not blurry.

> 3.In Section 3.5, the method of how hydrometeorological variables are compiled is not explicitly mentioned. Are these variables compiled directly from available datasets as listed in Table 5? Are there any newly generated time series estimates from existing data by, such as, model outputs or statistical calculations?

The time series estimates within GHI are based on existing observations (streamflow) and grid-based products (e.g., IMD and ERA5-Land). Using the newly created catchment boundaries, grid-based products were aggregated to the catchment scale using an area-weighted procedure. The time series of such aggregated values is one of the outputs from GHI. Section 3.5 was revised to make the reader better understand the output created by GHI. Moreover, a new graphic has been added in the Appendix (Figure A6) to describe the creation of area-weighted hydrometeorological data.

> 4.Some of the citation formats are incorrect. I see many places having the citation ends with double right parenthesis. For example, in Line 43, "River Discharge Data Set (RivDIS, Vorosmarty et al. (1998))". It can be written as "River Discharge Data Set (RivDIS) (Vorosmarty et al., 1998)". There are many other places with this issue (i.e., Lines 44, 54, 59, 115, 190, 242, 248, etc.). Please check throughout the manuscript.

As suggested by the reviewer, the double right parentheses issue was corrected in the revised manuscript, and relevant citations were revised accordingly. The revised manuscript was checked to make sure there are no occurrences of right or left double parentheses.